# Quercetin Administration Suppresses the Cytokine Storm in Myeloid and Plasmacytoid Dendritic Cells

**DOI:** 10.3390/ijms22158349

**Published:** 2021-08-03

**Authors:** Giulio Verna, Marina Liso, Elisabetta Cavalcanti, Giusy Bianco, Veronica Di Sarno, Angelo Santino, Pietro Campiglia, Marcello Chieppa

**Affiliations:** 1Department of Pharmacy, University of Salerno, 84084 Fisciano, Italy; gverna@unisa.it (G.V.); vdisarno@unisa.it (V.D.S.); 2National Institute of Gastroenterology “S. de Bellis”, Institute of Research, 70013 Castellana Grotte, Italy; marinaliso@libero.it (M.L.); elisabetta.cavalcanti@irccsdebellis.it (E.C.); giusy.bianco@irccsdebellis.it (G.B.); 3Unit of Lecce, Institute of Sciences of Food Production C.N.R., 73100 Lecce, Italy; angelo.santino@ispa.cnr.it

**Keywords:** quercetin, dendritic cells, plasmacytoid, cytokines, slpi

## Abstract

Dendritic cells (DCs) can be divided by lineage into myeloid dendritic cells (mDCs) and plasmacytoid dendritic cells (pDCs). They both are present in mucosal tissues and regulate the immune response by secreting chemokines and cytokines. Inflammatory bowel diseases (IBDs) are characterized by a leaky intestinal barrier and the consequent translocation of bacterial lipopolysaccharide (LPS) to the basolateral side. This results in DCs activation, but the response of pDCs is still poorly characterized. In the present study, we compared mDCs and pDCs responses to LPS administration. We present a broad panel of DCs secreted factors, including cytokines, chemokines, and growth factors. Our recent studies demonstrated the anti-inflammatory effects of quercetin administration, but to date, there is no evidence about quercetin’s effects on pDCs. The results of the present study demonstrate that pDCs can respond to LPS and that quercetin exposure modulates soluble factors release through the same molecular pathway used by mDCs (*Slpi*, *Hmox1,* and *AP-1*).

## 1. Introduction

Microbial or microbial component translocation through the intestinal mucosa can result in dendritic cell (DC) activation and inflammatory cytokines secretion [1]; this response needs to be limited to facilitate tissue repair and a return to intestinal homeostasis; if prolonged, chronic inflammation may arise and progress toward inflammatory bowel diseases (IBDs) [2,3,4]. Therefore, DCs residing in the gut-associated lymphoid tissues (GALT) are involved in the maintenance of tolerance toward the commensal microbiota that has a pivotal role in the development of IBDs [5]. These include ulcerative colitis and Crohn’s disease, which are characterized by epithelial barrier dysfunction [6] that may expose mucosal resident DCs to abundant luminal-derived lipopolysaccharide (LPS) and consequent excessive activation of immune cells besides the secretion of pro-inflammatory cytokines [7].

DCs can be divided into many subsets according to membrane proteins and their transcriptome [8,9]. By lineage derivation, two populations of DCs can be distinguished: myeloid (mDCs) and plasmacytoid DCs (pDCs) [10,11]. The former are derived from myeloid progenitors that are present in peripheral organs and act as antigen-presenting cells, they secrete pro- and anti-inflammatory cytokines to answer to external stimuli and drive other immune cells toward an effector [12] or tolerogenic response [13]. On the other hand, pDCs derive from lymphoid progenitors and are involved in the production of type I and III interferons (IFN); they are mainly involved in the defense against viruses but can also present antigens to initiate the adaptive immune response [14].

pDCs can be distinguished from mDCs by some specific membrane markers. Murine pDCs express less CD11c but, differently from mDCs, they are positive to B220 and Ly6C, thus appearing CD11c^int^B220^+^Ly6C^+^ compared to mDCs that are CD11c^high^B220^-^Ly6C^-^ [15]. Moreover, pDCs express fewer major histocompatibility complex II (MHCII) molecules than their myeloid counterparts [14].

pDCs are present in many districts of our body, including the gastrointestinal tract, they take part in oral tolerance and in the detection of potential pathogenic antigens (mostly exogen nucleic acids) that pass by Peyer’s patches (PP) and colonic mucosa. Once activated, they present antigens to effector cells in the mesenteric lymph nodes (MLNs) [16]. pDCs have an important role in IBDs, too [5,17]. There is evidence of their activity in the pathogenesis of IBDs; mice depleted of pDCs showed minor signs of inflammation and reduced symptoms of colitis [18,19], indicating a supporting role of pDCs in the onset of the inflammation. In addition, pDCs secretion of type I IFN is supposed to induce a protective and tolerogenic response in effector immune cells and thus reduce IBDs symptoms [20,21]. This dual role of pDCs in intestinal inflammation reflects their important role as immunomodulatory cells.

Based on our previous studies, demonstrating quercetin’s ability to suppress several pathways of the mDCs inflammatory response, in the present study, we focused our attention on the potential anti-inflammatory and antioxidant effects of pDCs exposure to quercetin. Quercetin is present in many foods (groceries and fruits) that are often called “superfoods” [22]. Together with several other groups, we demonstrated that quercetin could prevent bone marrow-derived DCs (BMDCs) inflammatory response to LPS administration, and, at the same time, promote tissue repair both in vivo and in vitro [23,24,25,26,27]. In particular, quercetin pre-exposure reduces DCs antigen-presenting activity and cytokine secretion by activating *Slpi* and chelating extracellular iron that is crucial in the activity and maturation of immune cells [24,26,28]. We have also previously observed amelioration of the inflammatory symptoms and dysbiosis in mice treated with a tomato-enriched diet, which is characterized by a high content of polyphenols, including quercetin [29,30]. Our data support a potential role for quercetin-enriched nutritional regimes as a preventive or therapeutic treatment of undesired intestinal inflammatory syndromes.

Despite the vast literature demonstrating the anti-inflammatory effects of quercetin and other polyphenols on immune cells, very little has been explored about the effect of quercetin administration on pDCs. However, quercetin and polyphenols influence the intestinal microenvironment and can protect from dysbiosis and inflammation, impacting DCs activity, too [29,31].

In the present study, we firstly addressed pDCs abundance in murine GALT, including PP and MLNs, isolated from wild-type (WT) mice and a model of spontaneous colitis (Winnie) [32]. Since we observed a significant presence of pDCs in WT and Winnie GALT, we sought to investigate if quercetin exposure could prevent pDCs inflammatory response and impair their maturation, particularly when treated with LPS.

Hence, we differentiated in parallel pDCs and mDCs (used as control) from murine bone marrow precursors and compared their cytokine secretion and the expression of selected genes previously identified as a key mediator in quercetin-induced anti-inflammatory response [23,26]. Our results demonstrate that pDCs respond to LPS administration, secreting a distinctive inflammatory cytokine repertoire; quercetin exposure suppresses LPS-mediated inflammatory response in a *Slpi*-, *Hmox1*-, and *Ap-1*-mediated but *Ptger2-*independent manner, confirming the potential relevance of quercetin administration as an adjuvant strategy for IBD patients.

## 2. Results

### 2.1. Differential Presence of Dendritic Cells in PP and MLNs of Winnie Mice

Intestinal inflammation often occurs when bacteria and bacterial antigens such as LPS penetrate a looser mucus layer and get closer to immune cells residing in the mucosa [33]; thus, we tested for their presence in PP and MLNs of WT and Winnie mice. FACS analysis revealed a similar abundance of CD11c^+^ cells in the PP as well as in MLNs of WT and Winnie mice (Figure 1A,B,D,E, respectively). When gated on CD11c^+^ cells, FACS data revealed that B220^+^ pDCs and B220^-^ conventional dendritic cells (cDCs) presence was similar in WT and Winnie PP (Figure 1C), but we observed a significant increase in CD11c^+^B220^-^CD8^-^ monocyte-derived dendritic cells (moDCs) in Winnie MLNs compared to their WT counterparts (Figure 1D–F). Lastly, we observed a reduction of CD11c^+^B220^+^ cells, both as steady-state dendritic cells (CD11c^+^B220^+^CD8^+^) and pDCs (CD11c^+^B220^+^CD8^-^) in MLNs of inflamed mice (Figure 1D–F).

### 2.2. Generation of Murine pDC from Bone Marrow Precursors

As in Winnie mice, the mucus layer is consistently more labile than in WT mice, pDCs are likely exposed to external antigens with LPS being one of them. We next investigated whether in vitro cultured pDCs could express toll-like receptor 4 (*Tlr4*) and are able to respond to LPS.

We generated pDCs and mDCs in vitro from murine bone marrow common precursors, with FMS-like tyrosine kinase 3 ligand**** (Flt3L) and granulocyte-macrophage colony-stimulating factor (GM-CSF) + interleukin 4 (IL-4), respectively. Using cytofluorimetric analysis, we confirmed the efficient maturation of bone marrow progenitors into CD11c^+^B220^+^Ly6C^+^ cells grown with Flt3L in the culture medium. Figure 2A–E describe the different surface markers expression in cells cultured with GM-CSF+IL-4 (CD11c^+^B220^-^Ly6C^-^ cells) versus Flt3L ones (CD11c^+^B220^+^Ly6C^+^ cells). Of notice, cells cultured with Flt3L were approximately 20% CD11c^+^B220^neg^, 45% CD11c^+^B220^int^, and 25% CD11c^+^B220^high^ cells. Interestingly, 24 h after LPS stimulation, Flt3L cultured cells were 5% CD11c^+^B220^neg^, 35% CD11c^+^B220^int^, and 45% CD11c^+^B220^high^. These data indicate that even in the presence of heterogeneous cell culture, cells responded to LPS upregulating B220 surface expression.

pDCs can respond to LPS by upregulating the expression of MHCII, but not CD80 (Figure 2F–G). In line with the aforementioned results, we detected *Tlr4* expression in pDCs (Figure 2H) even if significantly lower than what was observed in mDCs.

### 2.3. Quercetin Suppresses Cytokine and Chemokine Secretion in Both mDCs and pDCs

Based on these observations, we next investigated if pre-exposure to quercetin could suppress pDCs ability to release inflammatory cytokine and chemokine similarly to what was previously observed in mDCs. In our previous studies, 25 µM of quercetin was able to suppress the secretion of pro-inflammatory cytokines in mDCs stimulated with LPS without inducing cell death [23]. Thus, we used identical conditions to test quercetin’s effects on pDCs.

In line with our observations, quercetin significantly lowered the upregulation of MHCII and CD80 in pDCs stimulated with LPS (Figure 3A,B). Furthermore, quercetin did not affect cell viability as 7-Aminoactinomycin D (7-AAD) positive cells were about 30% in all experimental conditions (Figure 3C).

We tested the release of 36 cytokines, chemokines, and growth factors by quercetin pre-exposed pDCs 24 h after the activation with LPS. Vehicle-treated pDCs and mDCs were used as control. A comprehensive panel of the effects of LPS administration on mDCs and pDCs is shown in Figure 4, Figure 5 and Figure 6. In line with what was previously described, quercetin administration was able to suppress the secretion of most of the inflammatory mediators shown in the panels. Of note, despite the reduced expression of *Tlr4*, pDCs responded to LPS stimulation by secreting higher amounts of IL-10, IL-12p70, IL-27, and tumor necrosis factor (TNF) if compared to mDCs (Figure 4). They also secrete similar amounts of IL-6 and great quantities of chemokines such as chemokine (C-C motif) ligand 3 (CCL3), CCL4, CCL5, C-X-C Motif Chemokine Ligand 1 (CXCL1), CXCL2, and CXCL5 (Figure 5). Even more dramatically than mDCs, pDCs downmodulated the secretion of each factor analyzed when exposed to quercetin 24 h before the stimulation with LPS. Moreover, despite pDCs being the primary IFNα secretory cells, this cytokine was not detected in pDCs culture medium after stimulation with LPS (Figure 4).

Looking at the secretion of growth factors, mDCs are the main producers of GM-CSF, granulocyte colony-stimulating factor (G-CSF), macrophage colony-stimulating factor (M-CSF), and leukemia inhibitory factor (LIF); instead, pDCs secreted only a small amount of G-CSF, similar to mDCs after LPS administration (Figure 6).

### 2.4. Quercetin Modulates Inflammatory Molecular Signatures in pDCs

Previous results demonstrate that pDCs respond to LPS similarly to mDCs. Analyzing the mRNA expression of both cell types, we confirmed the expression of *Tlr4* by mDCs and, albeit lower, from pDCs (Figure 2H and Figure 7).

We have also recently demonstrated that quercetin affects intracellular signaling pathways that lead to the secretion of pro-inflammatory cytokines, by upmodulating the transcription of secretory leucoprotease inhibitor (*Slpi*) and heme-oxygenase 1 (*Hmox1*) genes, the latter involved in the reduction of oxidative stress.

Using RT-qPCR, we analyzed the expression pathways in our experimental conditions. As expected, *Slpi* and *Hmox1* expressions were induced by quercetin in mDCs, while their expression was lowered in cells stimulated solely with LPS. On the same note, pDCs showed similar *Slpi* and *Hmox1* transcription modulation induced by quercetin and LPS (Figure 7).

We further examined the expression of interferon regulatory factor 7 (*Irf7*) and *Irf8* genes, which are both involved in the activation of downstream effector proteins of the inflammatory response and the regulation of the secretion of inflammatory cytokines. As expected, both genes were expressed in pDCs, to a lesser extent than in mDCs, and yet they are downmodulated by quercetin, even in the presence of LPS.

Genes that regulate and modulate the inflammatory response and reduce oxidative stress, such as indoleamine 2,3-dioxygenase 2 (*Ido2*) and nuclear factor erythroid 2-related factor 2 (*Nrf2*), showed an increased expression in both mDCs and pDCs stimulated with LPS; however, *Ido2* levels are significantly lower in pDCs. Quercetin administration was able to reduce the cellular inflammatory state, as demonstrated by the significant reduction of the expression levels of both these genes. The molecular pathway induced by quercetin administration was independent of prostaglandin E receptor 2 (*Ptger2*) upregulation differently from what previously reported in macrophages [34], but it involved a marked increase in the expression of activator protein 1 (*Ap-1*) mRNA, the transcription factor that regulates gene expression in response to LPS, in both cell types exposed to quercetin; only in mDCs, its expression gets significantly modulated when quercetin is used to counter the stimulation with LPS.

## 3. Discussion

During the last few years, we have observed several anti-inflammatory effects induced by quercetin administration both in vitro using mDCs [23,24,25,26] and in vivo using acute and mild ulcerative colitis murine models [29]. Together with a vast literature [35], our results suggested that quercetin can act on multiple cell types with diverse molecular mechanisms. As pDCs are important but still poorly characterized players of the mucosa immune response, we wanted to understand whether quercetin exposure could suppress the inflammatory pathway and the maturation of pDCs.

pDCs have important roles in the immune response; they coordinate inflammatory responses driven toward viral infections as well as against bacteria [36]. They usually get activated by exogenous nucleic acids via TLR7 and TLR9 [37], but the response to LPS is still debated [38,39]. Despite that, in the present manuscript, we demonstrate that pDCs derived from murine bone marrow progenitors in vitro cultured with Flt3L can be activated and efficiently respond to LPS stimulation by upregulating the expression of surface markers and by secreting a variety of inflammatory cytokines.

Our research firstly demonstrates that pDCs represent a significant part of the CD11c^+^ population in murine PP and MLNs in physiologic and pathologic conditions. Compared to WT control, we observed an increase of moDCs and decreased steady-state DCs [40,41] that may reflect the inflammatory state of MLNs during active, but mild, colitis [32].

As pDCs are abundant in the GALT of WT and Winnie mice, we investigated whether quercetin pre-exposure could affect the inflammatory response of in vitro cultured pDCs similarly to what we observed with mDCs.

Flt3L is a growth factor implied in the selection and differentiation of bone marrow progenitors; it directs them toward a DC phenotype, and it is crucial in the generation of pDCs [14,42]. Using Flt3L we successfully generated pDCs after 10 days of culture and confirmed the expression of B220 and Ly6C surface proteins in more than 75% of CD11c^+^ cells.

pDCs respond to exogen nucleic acids, but during inflammation of the intestinal tract, the epithelial barrier becomes more permissive to bacterial product translocation, including LPS, which can further exacerbate the inflammatory responses [43]. In the present study, LPS was used as an inflammatory stimulus for pDCs. Response to LPS most commonly requires *Tlr4* expression [44,45]. The literature is not clear and often discordant about the expression of *Tlr4* by pDCs; however, in our experimental setup, Flt3L cultured cells generate a heterogeneous population enriched in pDCs that can respond to LPS and secrete inflammatory cytokines. Cell heterogeneity is further underlined by a low but still detectable concentration of IL-17A in the supernatants of mDCs and pDCs after LPS stimulation. Future studies will require cell sorting to improve cell purity and prevent these contaminations. Flt3L cultured cells express lower but still detectable *Tlr4* mRNA. In these experimental conditions, we cannot exclude that the detected *Tlr4* is expressed by the 20% of CD11c^+^B220^neg^ cells observed; nonetheless, surface B220 upregulation and cytokine secretion strongly suggest that pDCs efficiently respond to LPS administration. Moreover, MHCII expression is significantly upmodulated when pDCs are exposed to LPS.

Quercetin and polyphenols proved many times their anti-inflammatory and immunomodulating potential; however, little is known about their effects on pDCs. Here, we demonstrated that quercetin affects the inflammatory state of pDCs similarly to mDCs without substantially affecting cell vitality. Quercetin administration efficiently reduced the secretion of inflammatory cytokines from mature pDCs as well as mDCs. It also markedly prevented the overexpression of MHCII and CD80 following LPS administration. These observations imply that quercetin is also able to suppress the activation and maturation of pDCs.

In line with our previous observations on mDCs, the panel of 36 cytokines and chemokines provided useful insights on the inhibitory effects that quercetin has on inflammatory activity on DCs derived from different lineages. LPS activates the secretion of many cytokines in both cell types, with some cytokines differentially expressed, and quercetin can suppress their expression in mDCs as well as in pDCs. Looking at pDCs activated with LPS, our study shows surprisingly higher ability, even higher than mDCs, to secrete cytokines of the IL-12 family [46], in particular IL-12p70, IL-10, and IL-27 following LPS administration. Quercetin pre-exposure completely suppressed cytokine secretion in pDCs with an efficiency that appears superior to what was observed in mDCs. Similarly, the secretion of IL6, IL-18, and TNF is similar in LPS-stimulated pDCs and mDCs, but quercetin pre-exposure is significantly more efficient in suppressing pDCs cytokine secretion than mDCs. Some of the selected cytokines present in mDCs supernatants are not secreted by pDCs; these include IL-3, IL-4, IL-9, IL-13, IL-28, IL-31, IFNα, and IFNβ. With these results, we can assume that pDCs and mDCs perform non-redundant immunomodulatory activity after LPS stimulation, autoregulating their response to pathogens [47], but pDCs are more susceptible to quercetin pre-exposure. As mentioned before, Flt3L culture generated a heterogeneous cell population; thus, we wondered if the elevated concentration of selected cytokines detected in LPS-treated cells cultured with Flt3L could be the result of a synergistic stimulation between cDCs and pDCs. To address this question, we cocultured mDCs and pDCs to a 1:1 ratio and collected their supernatants 24 h after LPS stimulation. The results shown in Appendix A indicate that there was no synergistic effect induced by coculturing both cell types as cytokine concentration was not increased compared to mDCs and pDCs cultured alone.

To our surprise, pDCs stimulated with LPS did not secrete IFNα. That is indeed what was observed by Okada as LPS and TLR4 ligands did not activate IFNα secretion by DCs [48,49]. Appendix A shows that pDCs express *Ifnα2* when activated by a TLR9 agonist through the IRF7 pathway [50].

Quercetin pre-exposure completely prevents chemokine released by pDCs following LPS stimulation, apart from CXCL2 that is detectable, even if significantly reduced. Once again, quercetin pre-exposure effects are stronger in pDCs than mDCs.

The analysis of growth factors in both cell types displayed a marked difference in the secretion of G-CSF, M-CSF, GM-CSF, and LIF. To our knowledge, these factors induce the differentiation and the recruitment of neutrophils, macrophages, and other DCs; interestingly, there is evidence of a bias toward the differentiation of cDCs at the expense of pDCs when these factors are secreted at the site of inflammation [51,52,53,54]. Our data further demonstrate that mDCs recognize the threat of bacterial infection when stimulated with LPS and thus drive the immune response toward the recruitment of cells able to efficiently fight bacterial pathogens, while pDCs are more equipped to face viruses. In particular, the production of LIF is associated with the blocking of the differentiation toward pDCs, possibly explaining the non-significant reduced percentage of pDCs observed in Winnie MLNs [54]. Altogether, this evidence supports the results obtained in Winnie mice.

Then, we compared the molecular inhibitor pathway activated by quercetin. We know that the *Slpi* gene is pivotal for controlling this molecular pathway, as Slpi-KO DCs fail to respond to quercetin exposure [25]. As we expected, *Slpi* expression was upmodulated by quercetin alone and when it was used to suppress the effects of LPS in both mDCs and pDCs. mDCs and pDCs express *Irf7* and *Irf8*, which are genes that regulate the expression of pro-inflammatory cytokines; LPS stimulation induced an upmodulation of both these genes. However, pDCs express significantly less *Irf8*, which is involved mainly in the production of type I IFN after viral infections. As confirmed by the cytokine expression of IFNα, we can deduce that LPS stimulation does not activate a sustained antiviral response in mDCs and pDCs.

Quercetin administration induced the expression of *Hmox1* and *Ap-1* [55], which are two genes involved in the reduction of oxidative stress and the secretion of inflammatory cytokines. Since quercetin increases *Slpi* activity and suppresses nuclear factor kappa-light-chain-enhancer of activated B cells (NFkB), we can propose and support our hypothesis, as NFkB and *Ap-1* act alternatively in the activation of downstream pro-inflammatory cytokine genes. Moreover, if *Ap-1* gets expressed when NFkB is suppressed, cells undergo a process of progressive inactivation and might initiate apoptosis processes [56]. Even though quercetin suppresses oxidative stress and cellular inflammatory response, *Nrf2* expression is lower in cells exposed to this polyphenol. Many studies support the thesis that *Nrf2* is overexpressed during oxidative stress and inflammation and gets upmodulated by quercetin, albeit this takes place in the initial stages after quercetin administration [57]. In the present study, we observed the downstream effect of its activation, with the reduction of the cellular inflammatory state and the expression of the *Nrf2*-related gene *Hmox1*. Lastly, *Ido2* modulation in both mDCs and pDCs is concordant to its role as a regulator of the inflammatory response; this gene indeed acts as a controller of cytokine secretion and “forbids” immune cells to secrete excessive amounts of cytokines, thus driving uncontrolled inflammatory responses [58,59].

## 4. Materials and Methods

### 4.1. Animal Studies

Our investigations were performed under the relevant animal protocol, which was approved by the Institutional Animal Care Committee of National Institute of Gastroenterology “S. de Bellis” (Organism engaged for compliance of Animal Wellbeing: OPBA). All of the animal experiments were carried out according to the national guidelines of Italian Directive n. 26/2014 and approved by the Italian Animal Ethics Committee of the Ministry of Health—General Directorate of Animal Health and Veterinary Drugs (DGSAF- Prot. 768/2015-PR 27/07/2015). All animals were maintained in a controlled environment (20–22 °C, 12 h light and 12 h dark cycles, and 45–55% relative humidity).

### 4.2. Single-Cell Isolation from Murine Mesenteric Lymphnodes and Peyer’s Patches

WT and Winnie mice were sacrificed, and their MLNs and PP were detached, cleaned from fat, and put in RPMI 1640 medium (Thermo Fisher Scientific, Waltham, MA, USA). To obtain a single-cell suspension, MLNs were smashed with DPBS 1X (Thermo Fisher Scientific, Waltham, MA, USA) + 0.5 mM EDTA (Thermo Fisher Scientific, Waltham, MA, USA) and passed through 40 µm cell strainers (Miltenyi Biotec, Bergisch Gladbach, Germany). The single-cell suspension from PP was obtained after digestion with collagenase type IV and DNase I (Sigma Aldrich, St. Louis, MO, USA) for 30 min at 37 °C on a rocking platform. The resulting single-cell suspension was pelleted by centrifugation, washed with DPBS 1X + 0.5 mM EDTA, and passed through 100 μm, 70 μm, and 30 μm cell strainers (Miltenyi Biotec, Bergisch Gladbach, Germany). Then, cell pellets were washed with DPBS 1X + 0.5% bovine serum albumin (BSA, Sigma-Aldrich, St. Louis, MO, USA) and labeled for cytofluorimetric analysis, according to the manufacturer’s instructions.

### 4.3. Generation of mDCs and pDCs from Murine Bone Marrow

Dendritic cells from murine bone marrow were generated from six- to eight-week-old WT mice; they were sacrificed, and their tibiae and femurs were flushed with 0.5 mM EDTA (Thermo Fisher Scientific, Waltham, MA, USA). Then, red blood cells were lysed with an ACK buffer (Thermo Fisher Scientific, Waltham, MA, USA). The obtained single-cell suspension was split for mDCs and pDCs generation. For mDCs generation, cells were plated in 10 mL dishes at the concentration of 1 × 10^6^ cells/mL in RPMI 1640 medium (Thermo Fisher Scientific, Waltham, MA, USA) supplemented with 10% heat-inactivated fetal bovine serum (FBS, Thermo Fisher Scientific, Waltham, MA, USA), 100 U/mL penicillin/streptomycin (Thermo Fisher Scientific, Waltham, MA, USA), 1% HEPES 1M (Thermo Fisher Scientific, Waltham, MA, USA), 1% non-essential aminoacids 100 mM (Thermo Fisher Scientific, Waltham, MA, USA), 1% L-glutamine (Thermo Fisher Scientific, Waltham, MA, USA), 1% sodium pyruvate 100 mM (Thermo Fisher Scientific, Waltham, MA, USA), 25 ng/mL mGM-CSF (Miltenyi Biotec, Bergisch Gladbach, Germany), 25 ng/mL mIL-4 (Miltenyi Biotec, Bergisch Gladbach, Germany) as previously done, and cultured at 37 °C in a humidified 5% CO_2_ atmosphere.

For pDCs generation, cells were plated in a 6-well culture plate at the concentration of 2 × 10^6^ cells/mL in RPMI 1640 medium (Thermo Fisher Scientific, Waltham, MA, USA) supplemented with 10% heat-inactivated FBS (Thermo Fisher Scientific, Waltham, MA, USA), 100 U/mL penicillin/streptomycin (Thermo Fisher Scientific, Waltham, MA, USA), 1% HEPES 1M (Thermo Fisher Scientific, Waltham, MA, USA), 1% non-essential aminoacids 100 mM (Thermo Fisher Scientific, Waltham, MA, USA), 1% L-glutamine (Thermo Fisher Scientific, Waltham, MA, USA), 1% sodium pyruvate 100 mM (Thermo Fisher Scientific, Waltham, MA, USA), 200 ng/mL mFlt3L (Miltenyi Biotec, Bergisch Gladbach, Germany), and cultured at 37 °C in a humidified 5% CO_2_ atmosphere.

On day 10 after isolation, all non-adherent cells were gently harvested and plated on a 6-well culture plate at the concentration of 2.5 × 10^6^ cells/mL for cytofluorimetric analysis. Then, 25 µM quercetin (Sigma-Aldrich, St Louis, MO, USA) was administered to both pDCs and mDCs; 24 h later, cells were stimulated with 1 µg/mL of Salmonella Typhimurium LPS (Sigma-Aldrich, St. Louis, MO, USA) or 3 µg/mL CpG ODN 2216 or 3 µg/mL CpG ODN 2243 Control (Miltenyi Biotec, Bergisch Gladbach, Germany). Then, 24 h after that, cells were harvested, while the supernatants were collected and stored for further analysis.

For the coculture experiments, mDCs and pDCs were harvested from their respective culture plates on day 10 after isolation and plated in a 1:1 ratio in RPMI 1640 medium (Thermo Fisher Scientific, Waltham, MA, USA) supplemented with 10% heat-inactivated FBS (Thermo Fisher Scientific, Waltham, MA, USA), 100 U/mL penicillin/streptomycin (Thermo Fisher Scientific, Waltham, MA, USA), 1% HEPES 1M (Thermo Fisher Scientific, Waltham, MA, USA), 1% non-essential aminoacids 100 mM (Thermo Fisher Scientific, Waltham, MA, USA), 1% L-glutamine (Thermo Fisher Scientific, Waltham, MA, USA), 1% sodium pyruvate 100 mM (Thermo Fisher Scientific, Waltham, MA, USA). Then, 24 h later, cells were stimulated with 1 µg/mL of Salmonella Typhimurium LPS (Sigma-Aldrich, St. Louis, MO, USA). Then, 24 h after that, cells were harvested, while the supernatants were collected and used for cytokine detection.

### 4.4. Cytofluorimetric Analysis

mDCs and pDCs were detached from the plates with DPBS 1X (Thermo Fisher Scientific, Waltham, MA, USA) + 0.5 mM EDTA (Thermo Fisher Scientific, Waltham, MA, USA). Then, cells were washed with DPBS 1X + 0.5% BSA (Sigma-Aldrich, St. Louis, MO, USA) and labeled with CD8a FITC, Ly6C FITC, CD11c APC-Vio770, CD45R(B220) PE-Vio770, CD11c PE, MHCII APC, CD80 FITC, and 7-AAD Staining solution (Miltenyi Biotec, Bergisch Gladbach, Germany). Flow Cytometer acquisition was performed using NAVIOS (Beckman Coulter, Brea, CA, USA). Flow cytometer analysis was performed using Kaluza Software 1.5 (Beckman Coulter, Brea, CA, USA).

### 4.5. Cytokine Secretion Analysis

Multiplex cytokine assay was performed using a Cytokine & Chemokine Convenience 36-Plex Mouse ProcartaPlex™ Panel 1A (Thermo Fisher Scientific, Waltham, MA, USA) on cell culture supernatants from 3 independent experiments.

Cytokine secretion of coculture supernatants was also analyzed using ELISA kits for IL-6, IL-12p70, and TNF (R&D Systems, Minneapolis, MN, USA) following the manufacturer’s instructions.

### 4.6. RNA Extraction and qPCR Analysis

mDCs and pDCs after 10 days of culture were stimulated with LPS for 6 h and then harvested in TRIzol^®^ (Thermo Fisher Scientific, Waltham, MA, USA); total RNA was isolated from those cells according to the manufacturer’s instructions. One µg was reverse transcribed using an iScript cDNA Synthesis kit (Biorad, Hercules, CA, USA) with random primers for cDNA synthesis. Gene expression was assessed using the following primers: *GAPDH* Mm99999915_g1, *Irf8* Mm00492567_m1, *Ap.1* Mm00495062_s1, *Ptger2* Mm00436051_m1, *Nrf2* Mm00477784_m1, *Irf7* Mm00516791_g1, *Ido2* Mm00524210_m1, *Hmox1* Mm00516005_m1, *Slpi* Mm00441530_g1, and *Ifna2* Mm00833961_s1 (Thermo Fisher Scientific, Waltham, MA, USA). Real-time analysis was performed on a CFX96 System (Biorad, Hercules, CA, USA), and relative expression was calculated using the ΔΔCt method. At least three different experiments were performed.

### 4.7. Statistical Analysis

Statistical analysis was performed using GraphPad Prism 8 software (GraphPad Software, San Diego, CA, USA). All data obtained from at least three independent experiments were expressed as means ± SEM. We evaluated statistical significance with the two-way ANOVA test following Dunnett’s or Sidak’s as post-test. Results were considered statistically significant at *p* < 0.05.

## 5. Conclusions

The effect of quercetin administration on pDCs was never previously explored. Our results represent a comprehensive map of the inflammatory response to LPS by mDCs and pDCs. Furthermore, we describe the modulatory effect of quercetin exposure, underlining differences and similarities among these two DCs lineages. Results obtained by the present study demonstrate that quercetin administration can impact the inflammatory pathway of mDCs and pDCs, in both cases via *Slpi*, *Hmox1,* and *Ap-1* upregulation.

Results obtained from WT and the murine model of ulcerative colitis Winnie demonstrate that pDCs are as abundant as mDCs in the GALT, suggesting that future studies should address the effect of drugs and adjuvant to this important DCs lineage during chronic inflammatory syndromes.

## Figures and Tables

**Figure 1 ijms-22-08349-f001:**
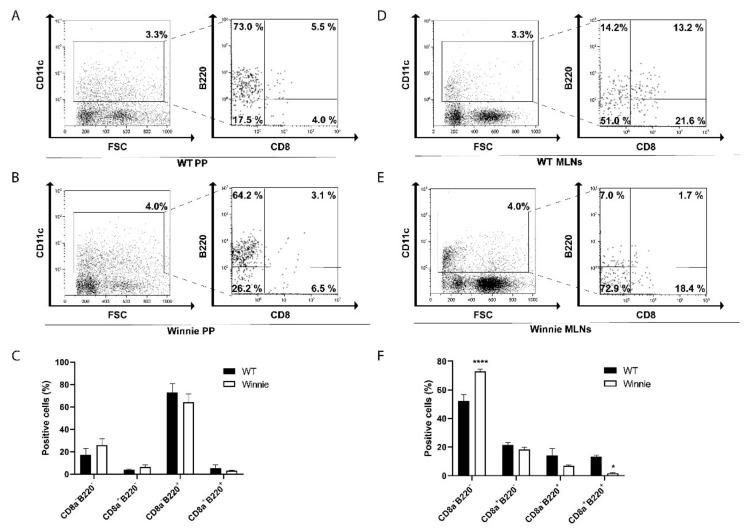
Representative density plots for dendritic cells (DCs) found in Peyer’s patches (PP) from wild-type (WT) (**A**) and Winnie mice (**B**), and in mesenteric lymphnodes (MLNs) from WT (**D**) and Winnie (**E**); 20,000 cells were acquired per each condition. Graphs represent percentages of the different populations of DCs in PP (**C**) and MLNs (**F**) in WT and Winnie mice; bars represent mean relative expression ± SEM (*n* = 4) for each genotype. * *p* < 0.05 ********
*p* < 0.0001.

**Figure 2 ijms-22-08349-f002:**
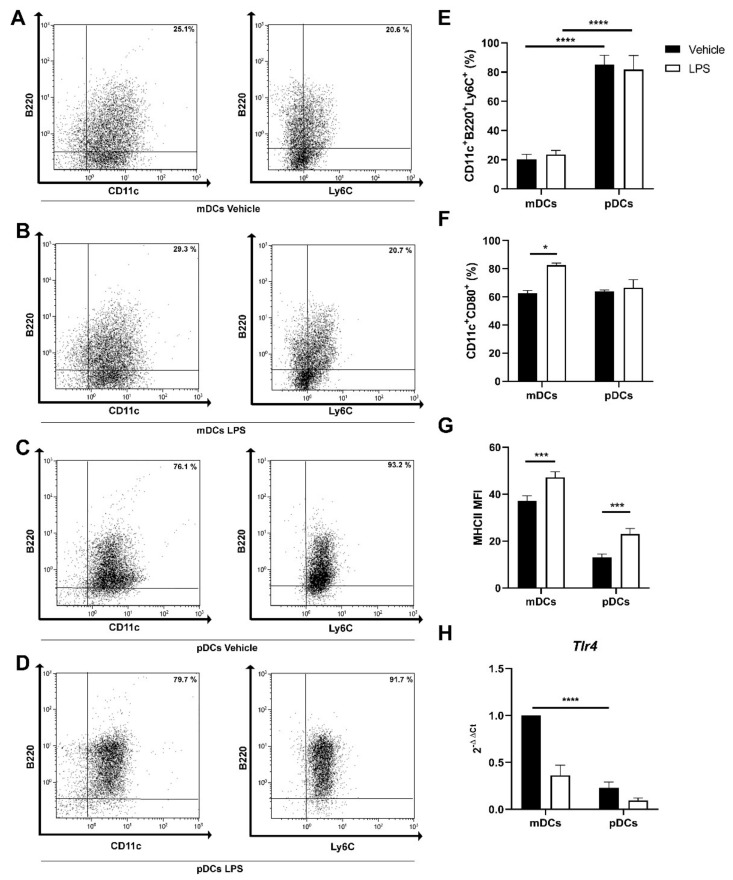
Representative density plots for CD11c, B220, and Ly6C staining in mDCs (**A**), myeloid dendritic cells (mDCs) stimulated with lipopolysaccharide (LPS) for 24 h (**B**), plasmacytoid dendritic cells (pDCs) (**C**), pDCs stimulated with LPS for 24 h (**D**). Bar plots for mean ± SEM (*n* = 4) showing the percentages of CD11c^+^B220^+^Ly6C^+^ cells (**E**), CD11c^+^CD80^+^ cells (**F**), and major histocompatibility complex II (MHCII) MFI (mean fluorescence intensity) (**G**) in control and after LPS stimulation for 24 h. Bar plots representing the mean ± SEM (*n* = 6) of toll-like receptor 4 (*Tlr4*) gene expression relative to control mDCs (**H**). * *p* < 0.05 *** *p* < 0.001 **** *p* < 0.0001.

**Figure 3 ijms-22-08349-f003:**
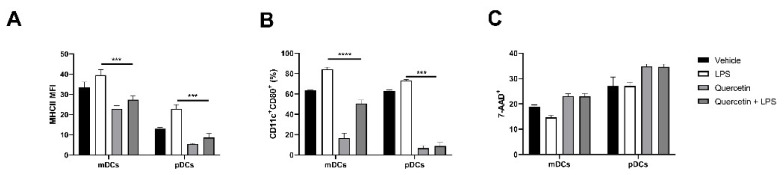
Bar plots expressing the mean ± SEM (*n* = 4) for MHCII MFI and CD80 expression (**A**,**B**) and 7-Aminoactinomycin D (7-AAD) vitality staining (**C**) of mDCs and pDCs in baseline conditions and after stimulation with LPS and/or quercetin. *** *p* < 0.001 **** *p* < 0.0001.

**Figure 4 ijms-22-08349-f004:**
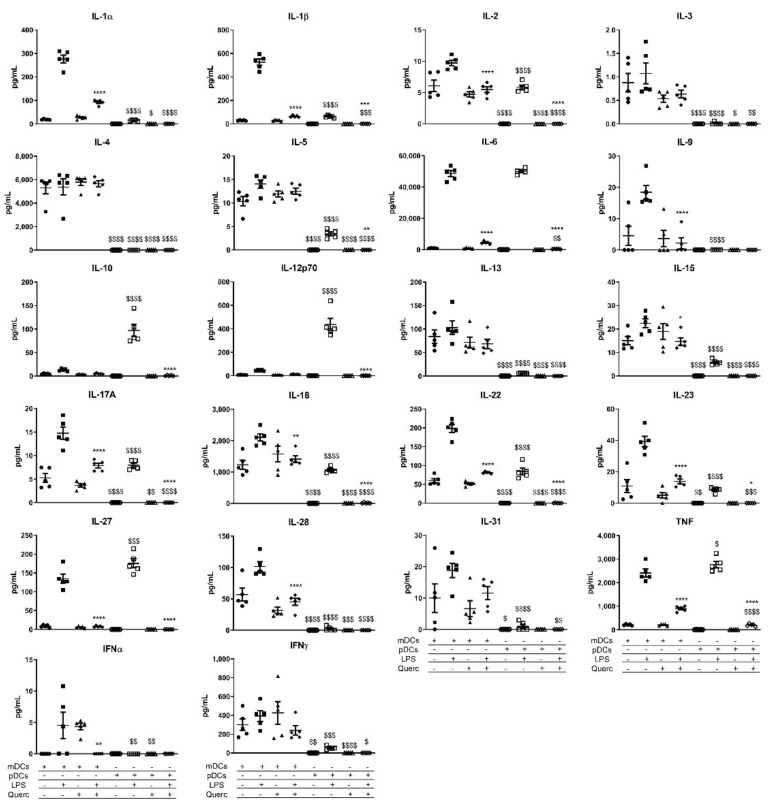
Scatter plots expressing the mean ± SEM (*n* = 5) for secreted cytokines of mDCs and pDCs in baseline conditions and after stimulation with LPS and/or quercetin. * *p* < 0.05 ** *p* < 0.005 *** *p* < 0.001 **** *p* < 0.0001; $ *p* < 0.05 $$ *p* < 0.005 $$$ *p* < 0.001 $$$$ *p* < 0.0001; * LPS vs. quercetin + LPS; $ mDCs vs. pDCs.

**Figure 5 ijms-22-08349-f005:**
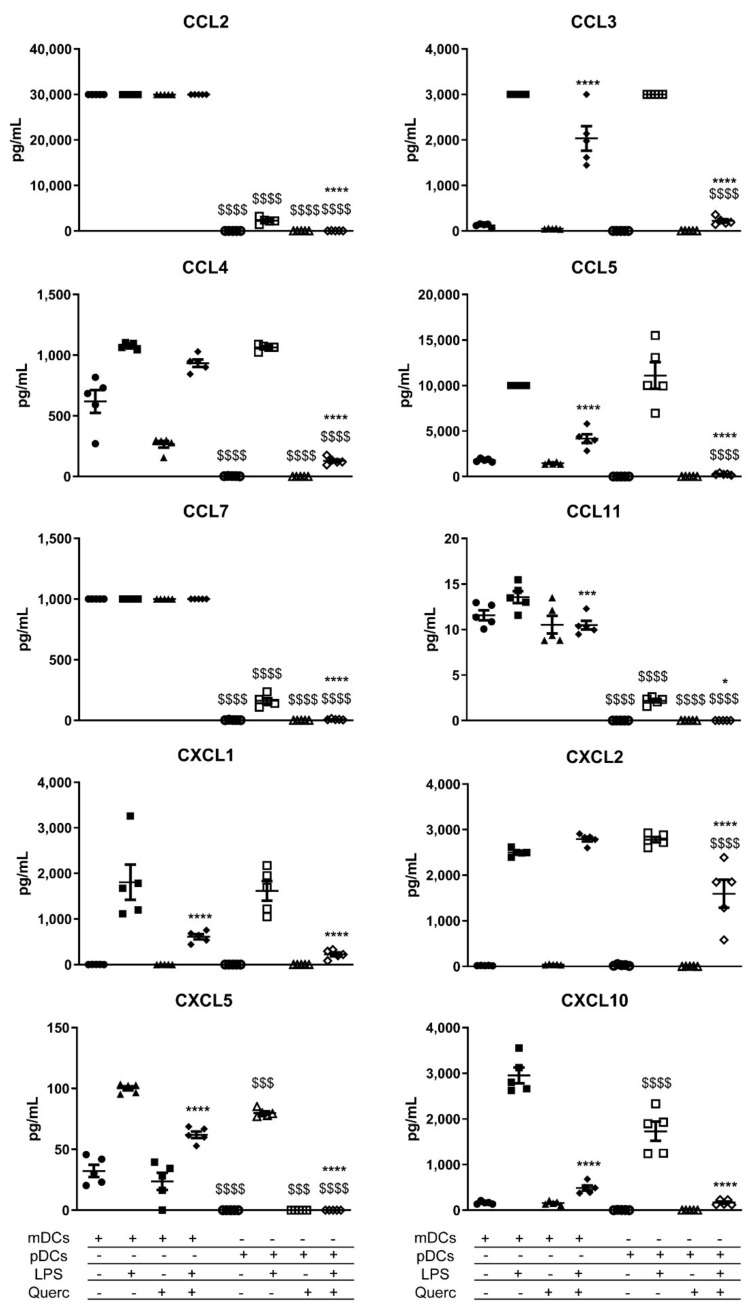
Scatter plots expressing the mean ± SEM (*n* = 5) for secreted chemokines of mDCs and pDCs in baseline conditions and after stimulation with LPS and/or quercetin. * *p* < 0.05 *** *p* < 0.001 **** *p* < 0.0001; $$$ *p* < 0.001 $$$$ *p* < 0.0001; * LPS vs. quercetin + LPS; $ mDCs vs. pDCs.

**Figure 6 ijms-22-08349-f006:**
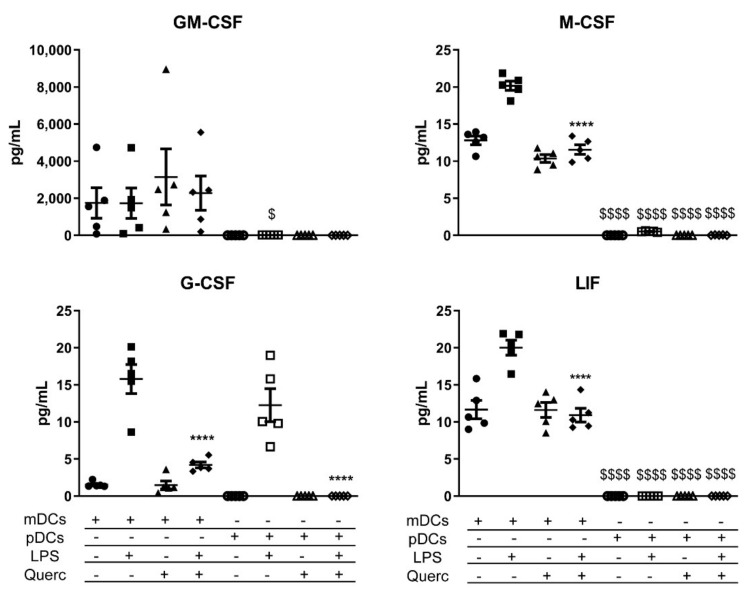
Scatter plots expressing the mean ± SEM (*n* = 5) for secreted growth factors of mDCs and pDCs in baseline conditions and after stimulation with LPS and/or quercetin. **** *p* < 0.0001; $ *p* < 0.05 $$$$ *p* < 0.0001; * LPS vs. quercetin + LPS; $ mDCs vs. pDCs.

**Figure 7 ijms-22-08349-f007:**
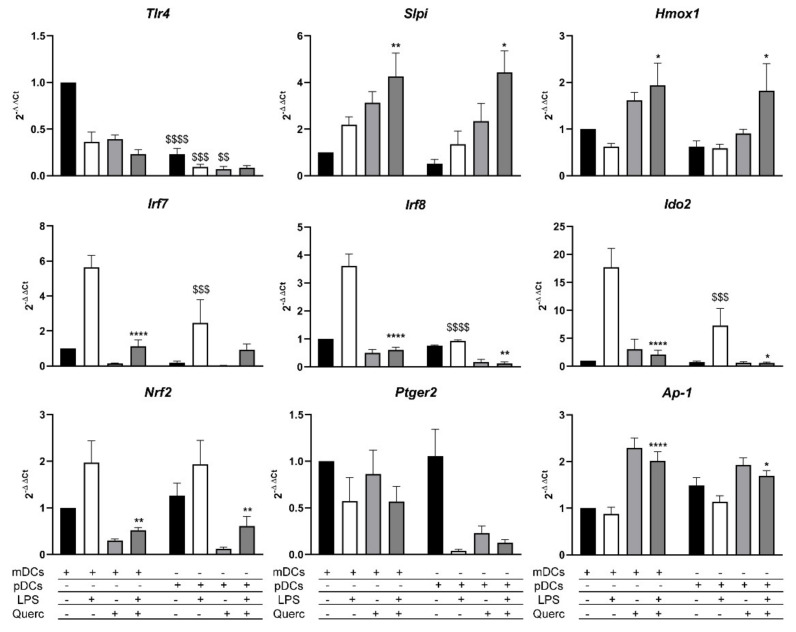
Molecular pathways activated in mDCs and pDCs 6 h after LPS stimulation and/or quercetin administration. Bar plots representing the mean ± SEM (*n* = 6) of each gene expression relative to control mDCs. * *p* < 0.05 ** *p* < 0.005 **** *p* < 0.0001; $$ *p* < 0.005 $$$ *p* < 0.001 $$$$ *p* < 0.0001; * LPS vs. quercetin + LPS; $ mDCs vs. pDCs.

## Data Availability

The data presented in this study are available on request from the corresponding author.

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
