# Peer review of "Quercetin Administration Suppresses the Cytokine Storm in Myeloid and Plasmacytoid Dendritic Cells"

_ijms, 2021, doi:10.3390/ijms22158349_

Round 1

Reviewer 1 Report

The authors answered the objections and resolved the critical issues. The new version of the manuscript is well written, the aim of the study is clearly perceived so the article in the present version is worthy for publication.

Author Response

We thank the Reviewer for his/her constructive comments that provided useful insights to improve this manuscript.

Reviewer 2 Report

In the manuscript entitled “Quercetin administration suppresses the cytokine storm in myeloid and plasmacytoid dendritic cells”, the authors investigated the modulatory effects of quercetin on pDCs stimulated with LPS. The authors found that in vitro generated pDCs express low levels of tlr4 and respond to LPS stimulation by up-regulating MHC-II molecules and secreting several pro-inflammatory cytokines. They show that in their in vitro culture model, qercetin inhibit the maturation of LPS-treated pDCs.

The following are some concerns:   

- In Figure 1, the authors should show the absolute number of positive cells, instead of the % of positive cells.

- Regarding the in vitro cellular model used by the authors to generate pDCs, I am surprised by the FACS profiles of the cells at the end of differentiation (Fig. 2C). On Fig.2C, it is shown that 76.1% of the cells are CD11c+ B220+, and are thus considered as pDCs. This is very surprising because this cell model generates roughly 2/3 of cDCs and at most 1/3 of pDCs (see for instance the initial article: Murine plasmacytoid pre-dendritic cells generated from Flt3 ligand-supplemented bone marrow cultures are immature APCs. Brawand et al, JI 2002). Whatever the yield, culturing BMDCs with FLT3L does not allow the generation of pure pDCs. This is an issue for the analysis of tlr4 expression since RT-PCR have been performed on a mix of cDCs and pDCs. Moreover, we cannot exclude that in this model, the LPS acts exclusively on the cDCs, which, once stimulated, produce molecules which will activate the pDCs in a second step. Only a cell sorting of pDCs at the end of the differentiation would make it possible to analyze the response of the pDCs to LPS.

- It is also very surprising to see that the LPS-stimulated pDCs secrete IL-17A (Fig. 4). Although the values ​​are low, the results seem to be very significant. How do the authors explain this? Once again, I suspect that this result illustrates the cellular heterogeneity of the model, which is not suited to answer the original question.

Author Response

In the manuscript entitled “Quercetin administration suppresses the cytokine storm in myeloid and plasmacytoid dendritic cells”, the authors investigated the modulatory effects of quercetin on pDCs stimulated with LPS. The authors found that in vitro generated pDCs express low levels of tlr4 and respond to LPS stimulation by up-regulating MHC-II molecules and secreting several pro-inflammatory cytokines. They show that in their in vitro culture model, qercetin inhibit the maturation of LPS-treated pDCs.

The following are some concerns:   

- In Figure 1, the authors should show the absolute number of positive cells, instead of the % of positive cells.

Figure 1 legend has been changed indicating that 20,000 cells were acquired for each sample.

- Regarding the in vitro cellular model used by the authors to generate pDCs, I am surprised by the FACS profiles of the cells at the end of differentiation (Fig. 2C). On Fig.2C, it is shown that 76.1% of the cells are CD11c+ B220+, and are thus considered as pDCs. This is very surprising because this cell model generates roughly 2/3 of cDCs and at most 1/3 of pDCs (see for instance the initial article: Murine plasmacytoid pre-dendritic cells generated from Flt3 ligand-supplemented bone marrow cultures are immature APCs. Brawand et al, JI 2002). Whatever the yield, culturing BMDCs with FLT3L does not allow the generation of pure pDCs. This is an issue for the analysis of tlr4 expression since RT-PCR have been performed on a mix of cDCs and pDCs. Moreover, we cannot exclude that in this model, the LPS acts exclusively on the cDCs, which, once stimulated, produce molecules which will activate the pDCs in a second step. Only a cell sorting of pDCs at the end of the differentiation would make it possible to analyze the response of the pDCs to LPS.

We agree with Reviewer 2 as we were surprised as well by the results obtained. We repeated the experiment several times as we were afraid to submit unreliable data, but the results were consistently the same.

As shown in Figure 2, in our culture with Flt3l we consistently obtained approximately 20% of B220neg cells, 45% of B220int and 25% B220high. Interestingly, 24 hours after LPS administration, among the CD11c+ cells, only 5% is B220neg cells, 35% is B220int and 45% B220high. A sentence has now been included in paragraph 2.2.

We agree that Tlr4 expression could be the result of contamination and that sorting is the only reliable strategy to exclude that, but data shown in Figures 4-5-6 indicate that this is not the case.

In particular: Flt3l cultured DCs produce a higher rate of IL-10 and IL-12 following LPS administration; this could not be the result of cDCs contamination. We were surprised by these data and investigated if the co-culture of 50% GM-CSF DCs and 50% Flt3l DCs could produce “adjuvant” effects for selected cytokine production. The results shown in Supplementary figure1 indicate that there was no synergistic effect induced by coculturing both cell types as cytokine concentration was not increased compared to mDCs and pDCs cultured alone.

We also addressed if Flt3l could efficiently respond to the conventional CpG administration. Supplementary figure2 indicates that CpG, but not LPS stimulated Flt3l DCs, increased the expression of Ifnα2.

- It is also very surprising to see that the LPS-stimulated pDCs secrete IL-17A (Fig. 4). Although the values ​​are low, the results seem to be very significant. How do the authors explain this? Once again, I suspect that this result illustrates the cellular heterogeneity of the model, which is not suited to answer the original question.

We agree with the Reviewer, the IL-17A is the result of cellular contamination that unfortunately, we are not able to exclude. A sentence in results and discussion has been included in the revised version to highlight this limitation.

Round 2

Reviewer 2 Report

Even if the methodology is insufficient to formally conclude, the authors made the effort to weight their conclusion in the revised manuscript.

This manuscript is a resubmission of an earlier submission. The following is a list of the peer review reports and author responses from that submission.

Round 1

Reviewer 1 Report

In this study, the authors first compared the population of mDCs and pDCs in Peyer’s Patches (PPs) and mesenteric lymph nodes (MLNs) in WT and Winnie mice. They identified increased monocyte-derived DCs and reduced steady-state DCs and pDCs in the MLNs in Winnie mice when compared with WT counterparts. The authors further characterized the cytokine and chemokine production from monocyte-derived mDCs and pDCs with LPS and demonstrated the anti-inflammatory effect of quercetin on LPS-stimulated mDCs and pDCs.

Although the authors demonstrated some interesting findings, there are several issues that should be addressed:

  1. While the authors demonstrated that Winnie mice exhibited altered monocyte-derived DC and pDC populations in PPs and MLNs when compared with their WT counterparts, they did not compare the cytokine and chemokine profiles of Winnie and WT mice. Authors should provide information on cytokine and chemokine profiles to correlate the animal results to downstream experiments, as well as demonstrate the role of pDCs in IBD.
  2. In Figure 2G, the authors examined the proportion of CD11c+B220+Ly6C+ DCs (pDCs) after stimulation of bone marrow derived macrophages with GM-CSF+IL-4 and Flt3L respectively. Although stimulated with Flt3L, the proportion of CD11c+B220+Ly6C+ DCs was only about 50%, suggesting the purity of the pDCs was not high, and this would affect the interpretation of downstream experiments. The authors should increase the purity of pDCs for the downstream cytokine and chemokine characterization. In addition, the authors should provide information on the proportion of mDCs after stimulation with GM-CSF+IL-4 for comparison.
  3. Although the authors showed reduction of cytokine, chemokine, and inflammation related genes after quercetin treatment, the authors did not explain how the dose of quercetin was selected. This information should be included in the article.
  4. The authors investigated and indicated regulation of inflammation and antioxidant related signaling molecules in mDCs and pDCs after stimulation with LPS and treatment with quercetin. However, for many the effects were not significant. The authors should perform additional experiments to support their stated conclusions.
  5. Figure 2F, Y-axis labels are missing. Please amend.

Reviewer 2 Report

This study describing the effect of Quercetin in the cytokines production of plasmacitoid dendritic cells is properly conducted but some points need to be clarify. The main problem of this manuscript is that it is not clear whether the authors want to focus the study's attention on the a) immunomodulatory effect of quercetin on DC (as assumed by the title) or b) on the comparison between mDC and pDC. This confusion is reflected throughout the entire paper and must be cleared up in order to make the article worthy for publication.

Specific comments

a) The format of the introduction could be improved by presenting a clear objective at the end of the introduction and by being more exhaustive in describing the mechanism that link pDC, IBD and quercetin; on the other hand, the basic description of DCs could be streamlined.

b) The procedure used to culture DC in presence of quercetin needs to be further precised:

  • why did the authors choose 25 µM quercetin for treatment? Are they based on previous data? In this case the reference must be mentioned.
  • Why did the authors decide to in vitro stimulate DC “first” with quercetin and “then” with LPS and not vice versa? My feeling is that this approach is more likely to be "preventive" than "therapeutic". According to the authors, can this treatment be translated as therapeutic?

c) Maturation is the key step in DC-mediated regulation of immune responses. Why the authors did not investigate whether LPS-induced DC maturation was attenuated by quercetin in terms of expression of, for example, class II MHC and costimulatory molecules, which are the main phenotypes of DC maturation, using flow cytometry? Have the authors investigate the effect of quercetin treatment on DCs in term of apoptosis/necrosis?

d) Regarding the effect of quercetin on cytokines secretion of LPS-stimulated DCs, the intent of the authors is unclear.

These results are not thorough enough in the discussion. The focus of the authors is not clear: does the project aim to a) study the effect of quercetin in the modular maturation of DCs induced by LPS or b) compare mDC and pDC? The second hypothesis is much less attractive to me.

Moreover in paragraph 2.3 of the results section, the authors indicate that IFNa is not detectable in the supernatants of LPS-stimulated pDC, even though this cytokine is secreted mainly by pDC. However, the data are not commented on in the discussion.

e) With reference to the comments at points a and d, the discussion must be reviewed keeping in mind the main purpose of the study, that must be highlighted.

Minor point:

f) In the Results section, paragraph 2.1, authors mentioned the “cDCs” subset: did they mean “Conventional” DCs?

g) Figure 2b is not of clear interpretation, not even in the electronic format. Authors should insert a clearer image, and, if possible highlight cells nuclei. I think that an image like this is not an added value for the manuscript and not of fundamental importance.